# Emerging Roles of YES1 in Cancer: The Putative Target in Drug Resistance

**DOI:** 10.3390/ijms25031450

**Published:** 2024-01-25

**Authors:** Eunjin Kook, Kyung-Soo Chun, Do-Hee Kim

**Affiliations:** 1Department of Chemistry, Kyonggi University, Suwon 16227, Republic of Korea; world1024@kyonggi.ac.kr; 2College of Pharmacy, Keimyung University, Daegu 42691, Republic of Korea; chunks@kmu.ac.kr

**Keywords:** Src family kinase, YES1, drug-resistance, cancer progression

## Abstract

Src family kinases (SFKs) are non-receptor tyrosine kinases that are recognized as proto-oncogenic products. Among SFKs, YES1 is frequently amplified and overexpressed in a variety of human tumors, including lung, breast, ovarian, and skin cancers. YES1 plays a pivotal role in promoting cell proliferation, survival, and invasiveness during tumor development. Recent findings indicate that YES1 expression and activation are associated with resistance to chemotherapeutic drugs and tyrosine kinase inhibitors in human malignancies. YES1 undergoes post-translational modifications, such as lipidation and nitrosylation, which can modulate its catalytic activity, subcellular localization, and binding affinity for substrate proteins. Therefore, we investigated the diverse mechanisms governing YES1 activation and its impact on critical intracellular signal transduction pathways. We emphasized the function of YES1 as a potential mechanism contributing to the anticancer drug resistance emergence.

## 1. Introduction

In spite of substantial progress and enhancements in cancer treatments over recent years, the occurrence of resistance to cancer therapies is still a common observation in clinical settings. In addition, despite substantial progress in understanding the molecular mechanisms of resistance to targeted therapies based on tyrosine kinase inhibitors, and in developing more effective drugs, advanced cancers remain incurable. This is because tumors eventually develop resistance and relapse [1]. Consequently, numerous researchers are focused on identifying novel molecular biomarkers that could predict treatment responses and prevent tumor relapse. It has been reported that Src family kinases (SFKs) play a role in advancing oncogenesis and in conferring resistance to chemotherapy drugs as well as to molecular targeted therapies [2].

SFKs are non-receptor tyrosine kinases that produce proto-oncogenes [3]. SFKs are recruited to the plasma membrane through interactions with transmembrane or adapter proteins, subsequently triggering intracellular signaling pathways that play roles in cell survival, adhesion, and motility [4]. At least nine SFKs are available in humans: Src, Yes, Lck, Fyn, Lyn, Hck, Blk, Fgr, and Yrk [4]. Within the SFK group, Yes, also known as YES1, shares the highest amino acid identity of 75% with Src, making it the closest paralog [5]. Src and YES1 serve as scaffolds for protein–protein interactions. Altered expression levels and catalytic activities of these proteins have been observed in various human tumors, including lung, breast, ovarian, gastric, and colon cancer [6,7]. Several reports have previously demonstrated the involvement of YES1 expression in cancer metastasis [8,9]. In the case of liver metastases, patients with elevated YES1 activity have shorter survival duration than those without YES1 activation [9]. In the later colon cancer progression stages, elevated YES1 levels and activity promote cell motility rather than tumor growth [8]. Recently, reports have focused on alterations in YES1 expression/activity concerning tumor growth, metastasis advancement, and resistance to targeted therapy, considering it a promising new target [5,10]. Alterations in YES1 activity are common in solid tumors, highlighting its role in oncogenic transformation, tumor growth, metastatic progression, and resistance to targeted therapies. In this context, we investigated various mechanisms driving YES1 activation and its influence on critical intracellular signal transduction pathways. We focus on YES1 function as a potential mechanism underlying the emergence of resistance to anticancer drugs.

## 2. Structure of YES1

SFKs are generally composed of six domain regions, which can vary based on their functions. One of these regions is the N-terminal Src homology (SH) 4 domain, which contains a critical sequence (Met-Gly-X-X-X-Ser/Thr) for the irreversible attachment of a myristate group as a lipid modification [11]. The SH3 domain is composed of 60 amino acid residues and facilitates protein–protein interactions by binding to proline-rich sequences [12]. The SH2 domain can bind to a specific tyrosine phosphorylation site with a conserved recognition pocket containing arginine. Additionally, it possesses a binding pocket for hydrophobic residues located C-terminally to the phosphotyrosine motif [13,14]. The Unique Domain (UD) of SFKs employs a mechanism to connect the disordered and structured domains within the SFKs [15]. Intermolecular autophosphorylation of a tyrosine residue located in the SH1 catalytic domain activation loop results in enhanced kinase activity [13]. C-terminal c-Src kinase (Csk) regulates the balance between the active and inactive conformations by facilitating the tyrosine phosphorylation of the negative regulatory tail, thereby inducing kinase activity autoinhibition [16]. In addition, within the kinase, there is a single Cys-277 residue situated on the P-loop that could potentially serve as a target for various covalent inhibitors [17]. Figure 1 shows the YES1 structure.

## 3. Interactions between YES1 and Intracellular Signaling Molecules in Cancer

Although the *YES1* gene is amplified in many tumors, SFK network deregulation can result from various other mechanisms. The following section summarizes the intracellular signaling molecules linked to YES1.

### 3.1. YES-Associated Protein 1 (YAP1)

YAP1, a YES1-associated protein, contains both a WW domain and a proline-rich (PXXP) motif that binds to the SH3 domain of YES1 [18]. YES1 activates YAP1 through phosphorylation, including known YES1 downstream signals, such as mitogen-activated protein kinase (MAPK) and Akt. YES1 and YAP1 interact within the context of β-catenin-active colon cancer SW480 cells. However, under the same conditions, binding between YAP1 and Src or Fyn could not be verified [19]. While both YES1 and Src can phosphorylate YAP1 at the Tyr-357 residue, only YES1 is crucial for the survival of β-catenin-active colon cancer cells [19]. β-catenin forms a ternary complex with the phosphorylated YAP1 as a transcriptional regulator and the transcription factor T-Box transcription factor 5 (TBX5) [19]. YES1 directs this complex to the anti-apoptotic gene promoters, such as Bcl-xL and survivin, thereby promoting colon cancer cell survival. This may be an important step in the malignant transformation progression [19]. Figure 2 outlines the signaling pathways described above.

### 3.2. Scribble (Scrib)

Scrib, as a multidomain polarity protein, is a member of the leucine-rich repeat and PDZ domain (LAP) protein family. It serves as a multifunctional scaffold for the assembly of diverse multiprotein complexes [20]. Scrib deregulation leads to mammary epithelial cell transformation, causing disruption in morphogenesis and cell polarity, and suppressing cell death [21]. When Snail is expressed in various cells, Scrib and β-catenin undergo mislocalization, leading to the interaction between Scrib and active YES1. This interaction results in YAP1 phosphorylation, which then translocates to the nucleus, thereby regulating cell polarity and growth signaling at cell–cell junctions [22]. In addition, Zhao et al. reported that wild-type Scrib interacts with YES1 in the inactive αC helix-out, adopting a closed conformation, while mislocalized Scrib interacts with active YES1 of the αC helix-in type, which is an open conformation [22].

### 3.3. Erythropoietin-Producing Hepatoma Receptor Tyrosine Kinase A2 (EphA2)

EphA2 is a member of the Eph family of receptor tyrosine kinases, which are involved in various biological and disease processes. The signaling mechanism of EphA2 involves lateral binding within the plasma membrane and contributes to various disease-related processes, including cancer and inflammation [23,24]. EphA2 promotes the epithelial–mesenchymal transition (EMT) of gastric cancer cells through the Wnt/β-catenin signaling pathway [25,26]. EphA2 overexpression led to the elevation of molecular markers associated with EMT, including N-cadherin and Snail, along with target activation within the Wnt/β-catenin pathway such as TCF4, Cyclin D1, and c-Myc [25]. In addition, EphA2 interacts with YES1, phosphorylating YES1 at the Tyr-426 residue [27]. Consequently, this interaction enhanced the proliferation and migration of gastric cancer cells. This effect is further demonstrated in a xenograft mouse model, where it promotes the growth and migration of gastric cancer [27]. Annexin 2 is a member of the annexin family, a group of calcium-mediated phospholipid-binding proteins. The phosphorylation of Tyr-24 in annexin 2 is closely linked to tumor invasion and metastasis due to its role in nuclear localization [28,29]. In connection with this report, the phosphorylation of annexin 2 at the Tyr-24 residue by YES1 led to its activation and an increase in nuclear distribution, which is associated with gastric cancer cell invasion and migration [27].

## 4. Post-Translational Modifications of YES1

Post-translational modifications (PTMs) in proteins, which include processes like enzymatic or chemical additions of covalent groups to amino acids, play a pivotal role in altering the physicochemical properties of these proteins. This leads to changes in their structure, localization, activity, and interaction with other molecular partners [30,31]. PTMs, notably ubiquitination, acetylation, glycosylation, phosphorylation, glutamylation, and methylation, are integral in the progression and development of cancer. Recent research has further highlighted the significance of these specific PTMs in contributing to the resistance against cancer therapeutics [32,33]. Sethi et al. observed a notable prevalence of mannose-type N-glycans in colorectal cancer tissues compared to the adjacent normal colon tissue. This modification alters the interaction between the drug and its receptor, serving as a biomarker for resistance to anti-EGFR therapy [34,35].

The N-terminal region of SFKs undergoes modification either through myristoylation alone or a combination of myristoylation and palmitoylation. Myristoylation involves the irreversible covalent attachment of myristic acid to the N-terminal glycine of a protein. N-myristoyltransferase (NMT1) facilitates the transfer of myristoyl groups to the terminal glycine or lysine residues of various proteins [36,37]. Myristoylation is a necessary step for attachment to the membrane, whereas palmitoylation assists in directing SFKs to distinct membrane microdomains [11]. It has been reported that myristoylation mediated by NMT1 results in abnormal oncogene Src signaling, which enhances the aggressiveness and proliferative ability of cancer cells, thereby contributing to the progression of cancer [38,39,40]. For ovarian cancer susceptible to drug resistance, the intersecting roles of NMT1 and Src signaling indicate that a new therapeutic strategy could involve the combination of NMT1 inhibitors with anticancer drugs [41]. It is possible that YES1 might also serve as a substrate for NMT [42]. The presence of myristoylation and palmitoylation in the N-terminal region of YES1 enables its specific localization to membranes rich in cholesterol and sphingolipids [43]. In COS-1 cells transfected with the YES1-expressing vector, YES1 accumulates in the Golgi region after biosynthesis and is subsequently transported to the plasma membrane via Cys-3 residue mono-palmitoylation [44]. The dual N-terminal acyl modification of YES1 leads to its preferential localization within the microdomain, which is related to its inability to induce cell transformation [45]. Therefore, the overlapping functions of NMT1 and YES1 signaling lead us to anticipate that new therapeutic strategies might involve combinations of NMT1 inhibitors and anticancer drugs.

Nitric oxide (NO) plays a crucial role in tumorigenesis, particularly in cancer cell invasion and metastasis. NO can reversibly bind to cysteine thiols, forming S-nitrosothiols that modulate the enzymatic activities of specific target proteins [46]. NO induces S-nitrosylation of Src at Cys-498, leading to the activation of its kinase activity. Additionally, the Cys-506 residue of YES1, which corresponds to Cys-498 of Src, serves as a crucial site for NO-mediated modification, potentially enhancing YES1 activity [47]. Src activation by NO is pivotal for cancer cell proliferation and invasion, potentially by inhibiting E-cadherin expression. These physiological phenomena induced by NO may be mediated through the activation of SFKs, including YES1 [47].

## 5. Role of YES1 Regulation in Tumor Development and Resistance to Cancer Drugs

Drug resistance in cancer can arise from various mechanisms, including changes in drug metabolism, alterations in drug targets, and the genetic adaptations in cells that bypass the pathways targeted by treatment [48]. The aspect of resistance that has been most thoroughly studied involves the metabolism of drugs, covering their uptake, efflux, and detoxification [49,50]. Mutations that either change the function or lower the expression of surface receptors and transporters can also lead to resistance [49,50]. In addition, oncogene addiction, where cancer cells become overly reliant on a specific oncogene, provides a basis for the development of targeted therapies [51,52]. For instance, resistance often arises due to mutations in the gatekeeper residues of the kinase domain, which hinder the binding of drugs [53,54]. Studies have shown that these resistance mutations can be present in small subpopulations of tumor cells, suggesting that targeted therapies should be selected for these mutant forms [55]. Moreover, cancer cells have been found to employ other methods to evade the impact of targeted inhibitors, such as the amplification of different oncogenes or the deactivation of other cell survival pathways [56]. It is expected that YES1 can be activated in the process of cancer drug resistance, and the activation serves as a compensatory mechanism for the signal transduction inhibited by target therapy. In this section, we will focus on the function and control of YES1 in both the development of cancer and the mechanisms of resistance to cancer drugs.

### 5.1. Skin Cancer

Since a long time ago, there have been reports of increased c-src proto-oncogene expression in melanoma [57]. Several studies have indicated that SFKs are activated in skin cancer, but there is controversy regarding the specific SFK types implicated in skin cancer progression [58,59]. In this context, we focused exclusively on YES1. The expression and kinase activity of YES1 were consistently higher in 20 human melanoma cell lines than that of melanocyte cell lines. In contrast, there was no significant difference in Src expression between these two cell lines [60]. A subcellular fractionation analysis revealed that YES1 is observed in the plasma membrane, perinuclear region, and cytoplasmic compartments, whereas Src is predominantly linked to the plasma membrane [60]. Loganzo et al. suggested that alterations in YES1 expression may be involved in the malignant advancement of human melanocytes [60]. In addition, YES1 kinase activity is increased in human brain metastatic melanoma cell lines exposed to neurotrophin and nerve growth factor, whereas Src remains unaffected [59]. YES1 is more important than Src in melanoma progression and metastasis. Cells with a high propensity for brain metastasis in melanoma show intrinsically elevated YES1 activity compared to their parent or less metastatic counterparts [59]. Moreover, neurotrophin seems to amplify YES1 activity, facilitating signal penetration into neurotropin-rich stromal microenvironments such as the brain [59]. Neurotrophins seem to enhance YES1 activity, enabling signal transmission within neurotrophin-rich stromal microenvironments such as the brain. Other reports have verified that YES1 is expressed at higher levels in squamous cell carcinoma than in other skin cancer types [58].

Somatic LKB1 mutations occur in 10% of cutaneous melanoma. Mice with melanocyte-specific Lkb1 loss and K-Ras activation have been shown to develop highly metastatic melanomas [61]. LKB1 deficiency results in the expansion of a highly invasive and tumor-clonogenic subpopulation of cells with elevated CD24 expression, which serves as a metastasis modulator and a stem-progenitor cell marker. The expansion of the CD24-positive subpopulation leads to an elevated YES1 phosphorylation and enhanced expression of Wnt target genes, subsequently promoting melanoma formation and metastasis compared to isogenic CD24-negative cells [61]. In addition, in melanoma A375 cells, YES1 overexpression triggered MEK-independent ERK activation. ERK activation is driven by YES1, suggesting a potential mechanism that contributes to drug resistance in patients with melanoma and BRAF mutations who undergo combined BRAF and MEK inhibition [62].

### 5.2. Breast Cancer

Breast cancer can be categorized into molecular subtypes according to gene expression patterns, including luminal-like, ErbB2+ (human epidermal growth factor receptor 2 (HER2)-enriched), and basal-like [63]. In identifiable subsets of basal-like and HER2+ breast cancers, Bilal et al. suggested that several Src proto-oncogenes, including Lyn, Yes, Hck, Fyn, and Lck, were overexpressed and could serve as promising therapeutic targets because of their high degree of connectivity in the tumor network [64]. YES1 exhibits relatively high expression levels in the HER2-overexpressed patient group, which is associated with a high recurrence rate [64]. The inhibition of YES1 expression using shRNA has been demonstrated to impair the growth of basal-like breast cancer cell lines, including MDA-MB-468, MDA-MB-231, and BT549 [64].

The gene amplification, overexpression, and activation of YES1 have been observed in trastuzumab-resistant HER2-positive breast cancer cells (BR-474-R) and trastuzumab/lapatinib-dual-resistant BT-474-RL2 cells. The pharmaceutical inhibition of YES1 using the Src inhibitor dasatinib effectively restored sensitivity to trastuzumab and lapatinib in both BT-474-R and BT-474-RL2 cells [65]. The combination treatment resulted in a greater inhibition of Akt and HER2 phosphorylation than when each treatment was administered individually [65]. In addition, the combination treatment induced G1 phase cell cycle arrest in the resistant cell lines [65]. Relapse-free survival and disease-specific survival of patients with HER2-positive breast cancer with higher YES1 mRNA expression were significantly shorter than those of patients with lower expression [65].

Neratinib is an oral pan-HER inhibitor that effectively and irreversibly inhibits tyrosine kinase activity [66]. YES1 amplification was detected in HER2-positive breast cancer BT-474 cells that had developed resistance to neratinib. YES1 binds to HER2 in neratinib-resistant breast cancer cells [67]. The sensitivity of these cells to neratinib was restored with YES1 knockdown using siRNA and pharmacological inhibition using dasatinib. Additionally, the combination of dasatinib and neratinib exhibited strong antitumor activity against YES1-amplified neratinib-resistant cells [67]. The combination of YES1 knockdown and neratinib treatment resulted in a more pronounced inhibition of the phosphorylation of HER2, AKT, and MAPK than either treatment alone [67].

Trastuzumab–emtansine (T-DM1) is an antibody–drug conjugate that targets human HER2-positive metastatic breast cancer. Moreover, it is effective against trastuzumab-resistant metastatic breast cancer [29]. YES1 is overexpressed in T-DM1-resistant cells because of the amplification of the chromosomal region 18p11.32, which contains the YES1 gene [68]. BT-474 cells with HER2 amplification that acquired resistance to trastuzumab–emtansine (BT-474-R/TDR) exhibited increased HER2 expression and Akt phosphorylation. The phosphorylation levels of Src, along with YES1 expression, were upregulated in BT-474-R/TDR cells [69]. YES1 knockdown or dasatinib administration is effective even after acquiring resistance to both trastuzumab and T-DM1 [69]. Therefore, a promising approach for overcoming resistance to HER2-targeted drugs involves the simultaneous inhibition of both HER2 and YES1.

### 5.3. Lung Cancer

Human lung adenocarcinoma comprises distinct subtypes of lung cancer, each of which exhibits unique cellular and mutational heterogeneities. Researchers have discovered epidermal growth factor receptor (EGFR) mutations in patients with lung adenocarcinoma, which have been linked to their response to EGFR inhibitors [70]. Third-generation EGFR-TKIs, such as osimertinib and rociletinib, have been developed specifically to target mutant forms of EGFR and overcome T790M-mediated resistance [71,72,73]. Similar to other EGFR TKIs, the response to osimertinib varies in both extent and duration, ultimately leading to the emergence of resistance. Osimertinib effectively suppressed EGFR phosphorylation, but partially inhibited the phosphorylation of key PI3K pathway components, such as PDK1, AKT, and S6. The phosphorylation of MAPK pathway components, including BRAF, MEK, and ERK, continues to reactivate following osimertinib treatment [74]. In addition, YES1 amplification was detected in osimertinib-resistant EGFR-mutant lung cancer (PC-9/BRc1/9291) cells [74]. Inhibiting YES1 through pharmacologic or genetic means restored osimertinib sensitivity in these cells. Moreover, concurrent dasatinib and osimertinib administration led to a decreased tumor growth rate in a xenograft model involving PC-9/BRc1/9291 cells [74]. Patients with EGFR-mutant NSCLC exhibit YES1 amplification upon developing resistance to EGFR-TKIs [75]. In addition, among patients with ALK fusion-positive lung cancer who had progressed on ALK TKIs, YES1 amplification was detected in 2 of 17 samples [75].

The overexpression of YES1 results in resistance to both EGFR and ALK TKIs. YES1 depletion in YAP1-overexpressing cells abolished YAP1 phosphorylation at Tyr-357 and fully restored sensitivity to EGFR or ALK TKIs [76]. EGFR-mutant and ALK fusion-positive cells overexpressing YES1 or YAP1 showed resistance to EGFR and ALK TKIs but remained sensitive to the combined inhibition of the primary driver and YES1 [76]. In a patient with stage IV lung adenocarcinoma and detectable YES1 amplification, treatment with dasatinib for 10 weeks resulted in a reduction in lung tumor size [76]. Similarly, the inhibition of tumor growth in patient-derived xenograft models using dasatinib depends on the YES1 status. Dasatinib selectively blocks the proliferation and invasion of high-YES1-expressing cell lines [77]. In addition, Minari et al. reported that YES1 amplification alone is not a sufficient indicator of ALK-TKI resistance [78]. A more specific analysis of whole exome sequencing suggested that YES1 and MYC amplification may be associated with resistance to ALK TKIs [78]. Incorporating dasatinib with brigatinib or lorlatinib restores sensitivity to ALK TKIs in MYC/YES-overexpressing H3122 lung cancer cells [78].

Recently, a novel YES1 kinase inhibitor, CH6953755, with high specificity for YES1, was recently developed. This compound induced responses in YES1-amplified esophageal and small-cell lung cancer (SCLC) xenograft models [79,80]. The disruption of YES1 signaling in SCLC models leads to a significant reduction in tumor growth and metastasis, indicating a potential reliance on this oncogenic pathway [80].

### 5.4. Leukemia

Several SFKs are expressed in acute myeloid leukemia (AML), and they have been reported to promote AML cell survival [81]. Homeobox (HOX) genes are key factors in leukemia development [82]. Meis homeobox 2 (MEIS2) was aberrantly expressed at high levels in patients with AML1-ETO-positive AML. MEIS2 promotes AML1-ETO-associated leukemogenesis by restricting the binding of a repressive AML1-ETO complex to proto-oncogenes. Elevated MEIS2 expression hinders the binding of AML1-ETO to the YES1 promoter, leading to the enhanced expression of this proto-oncogene in human AML cells. YES1 depletion leads to a significant decrease in cell proliferation and a remarkable reduction in colony formation in AML1-ETO-positive Kasumi cells [83]. Vegi et al. suggested that the synergistic effect of MEIS2 and AML1-ETO on leukemogenesis may be linked to kinases, suggesting an avenue for targeting this cooperation with approved drugs such as dasatinib [83]. In another study, an analysis of transcriptomes using microarrays in leukemia K562 cells resistant to nilotinib and imatinib demonstrated an increase in the expression of genes encoding kinases, including AURKC, Fyn, Syk, Btk, and Yes [84].

### 5.5. Liver Cancer

Hepatocellular carcinoma (HCC) exhibits decreased Csk levels compared to normal liver tissue, and this diminished expression is linked to increased Src activity [85]. YES1 is expressed not only in the membrane and cytoplasm, but also in the nuclei of cancer cells in human hepatocellular carcinoma tissues and a human HCC cell line. The nuclear localization of YES1 with CDK1 has been observed during the initial phases of hepatocarcinogenesis and in well-differentiated HCC, implying that the presence of YES1 within the nucleus could serve as a valuable indicator for detecting early-stage HCC [86]. Blocking the nuclear expression of YES1 may offer a new strategy for impeding HCC development [86].

### 5.6. Glioma

Given the broad influence of SFK signaling on key pathways in glioblastoma multiforme, the small-molecule inhibitor dasatinib has emerged as an innovative therapeutic option [87]. Fyn, Yes1, and Src were consistently expressed in both glioma stem cells and primary glioma cells among SFKs, whereas LCK was exclusively expressed in primary glioma cells [88]. Although dasatinib, an SFK inhibitor, effectively inhibited glioma stem cell migration, it did not affect the CD133-positive glioma stem cell proportion among the glioma spheres. This implies that YES1 is not involved in cell growth or self-renewal ability [88].

### 5.7. Sarcoma

Ewing sarcoma is an extremely aggressive cancer that affects both bones and soft tissues, making it the second most prevalent primary malignant bone tumor in children and adolescents. Indovina et al. suggested the potential application of targeted SFK inhibition in Ewing’s sarcoma treatment [89]. The significance of targeting SFKs in sarcomas has been increasingly emphasized. Malignant mesothelioma is a highly aggressive cancer that is unresponsive to current chemotherapy treatments [90]. A phosphotyrosine proteomic analysis indicated the activation of SFKs such as Src, Fyn, and YES1 in malignant mesothelioma [91]. Among these SFKs, only the inhibition of YES1 resulted in the suppression of cell growth in malignant mesothelioma cells. Disrupting YES1 leads to the inactivation of β-catenin signaling, subsequently reducing the levels of Cyclin D required for the G1-S transition in the cell cycle [92].

As a member of the Crk adaptor protein family, CrkL possesses an SH2 domain and two SH3 domains [93]. The overexpression of the adaptor protein Crk-like (CrkL) enhances the growth and progression of rhabdomyosarcoma [94]. The expression level of the CrkL protein is markedly high in cell lines, xenografts, and human tumor samples in both alveolar and embryonal subtypes of rhabdomyosarcoma [94]. In sarcoma Rh30 cells, the reduction in CrkL protein expression resulted in a decrease in the phosphorylation and activation of YES1, subsequently causing cell cycle arrest of rhabdomyosarcoma cells and delayed xenograft growth [94].

### 5.8. Ovarian Cancer

An activation of SKFs has been linked to their overexpression in late-stage human ovarian cancer. SFKs are potential targets for ovarian cancer treatment [95]. YES1 is overexpressed in tumor samples obtained from patients with breast and ovarian cancer, as well as in recurrent or resistant cases of ovarian cancer [96,97]. The phosphorylation of YES1 influences the expression of several cell cycle regulators. Multiple sites on the unique N-terminal domain of YES1 are phosphorylated by the cell cycle kinase CDK1 during mitotic arrest induced using anti-tubulin drugs [97]. YES1 plays a role in modulating sensitivity to anti-tubulin chemotherapy [97].

MicroRNAs (miRNAs), specifically miR-199a, play a significant role in cancer regulation [98,99]. These noncoding RNAs are approximately 22 nucleotides in length and function in post-transcriptional gene expression regulation [100]. In cisplatin-resistant ovarian cancer cell lines, there was a notable decrease in the expression of miR-133a. However, when these cells were treated with an miR-133a mimic, their resistance to cisplatin was significantly reduced. miR-133a functions by binding to the 3′-untranslated region (UTR) of YES1, leading to a down-regulation of its expression [101].

### 5.9. Prostate Cancer

miR-199a, particularly notable as a cancer inhibitor in various types of cancers, is notably down-regulated in prostate cancer [99]. This down-regulation is associated with resistance to paclitaxel in prostate cancer [99]. By targeting the 3′-UTR, miR-199a can inhibit the expression of YES1. This inhibition retards the growth of prostate cancer in mouse xenografts and helps overcome paclitaxel resistance by reducing YES1 expression [99]. Conversely, the increased expression of YES1 can negate the effects of miR-199a in resisting paclitaxel [99].

## 6. Concluding Remarks and Future Perspectives

The amplification of the YES1 gene, along with its increased expression and activation, has been observed in numerous malignancies. The emerging expression of YES1 in various cancer cells resistant to drugs has been increasingly reported in numerous studies, suggesting that it will be a notable target for future investigations. According to a clinical case study conducted in China, patients with NSCLC carrying an EGFR-sensitive mutation and YES1 amplification received afatinib treatment as first-line therapy. However, after four weeks, they noticed that disease was progressing again. Throughout the following rounds of chemotherapy, the patients’ disease progressed rapidly [102]. YES1 amplification could potentially correlate with primary resistance to EGFR-TKIs and may serve as a negative prognostic factor for EGFR-TKI therapy in patients with NSCLC carrying EGFR-sensitive mutations [102]. Clinical trials involving patients with lung cancer have validated the significance of YES1, indicating that it is likely to garner increasing attention. It is anticipated that modulating the expression or activity of YES1 offers a reliable approach and can serve as a complementary alternative to conventional chemotherapy. This therapeutic method could provide an additional or alternative option in the treatment of cancer by focusing on YES1 function modification in tumor cells. In this review, we explored different mechanisms that lead to the activation of YES1 and its impact on key intracellular signaling pathways. Our research primarily concentrated on the role of YES1 function as a possible factor contributing to the development of resistance to anticancer drugs.

YES1 serves as a factor that contributes to resistance against the EGFR inhibitors trastuzumab or neratinib, and its suppression using dasatinib can reverse this resistance [65,67,75]. However, dasatinib is a multi-targeted kinase inhibitor known for its ability to block Bcr/Abl, c-Kit, PDGFR, and SFK family members, including Src, Lck, Hck, Yes, and Fyn [103]. A few investigators are increasingly intrigued by the quest for inhibitors designed specifically for YES1. As explained previously, we believe that many researchers will continue to discover YES1-specific inhibitors such as CH6953755 in the future. In addition, we expect YES1 to emerge as a prominent regulatory target in SCLC, particularly in the absence of specific target molecules. Recently, Du et al. reported the possibility that the Cys277 residue of YES1 was targeted by covalent Src inhibitors [104]. Moreover, investigations are necessary to explore inhibitors targeting the cysteine residue of YES1, particularly in the context of drug-resistant cancers. Finally, we would like to highlight the importance of further research to investigate the connection with tumor resistance, focusing on uncovering the still unidentified regulatory mechanisms of YES1 in transcription, translation, and post-translational modification.

## Figures and Tables

**Figure 1 ijms-25-01450-f001:**
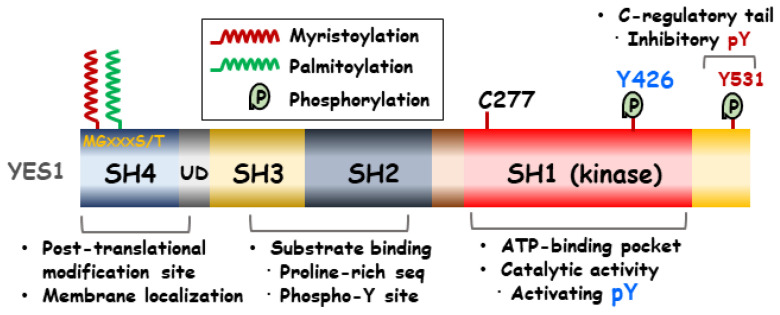
Functional domains of YES1. The YES1 protein makes up four functional domains. Among these, the N-terminal SH4 domain is involved in post-translational modification and membrane localization. The SH3 and SH2 domains are two key domains required for the protein–protein interaction. The SH1 kinase domain plays an important role in the catalytic activity of YES1.

**Figure 2 ijms-25-01450-f002:**
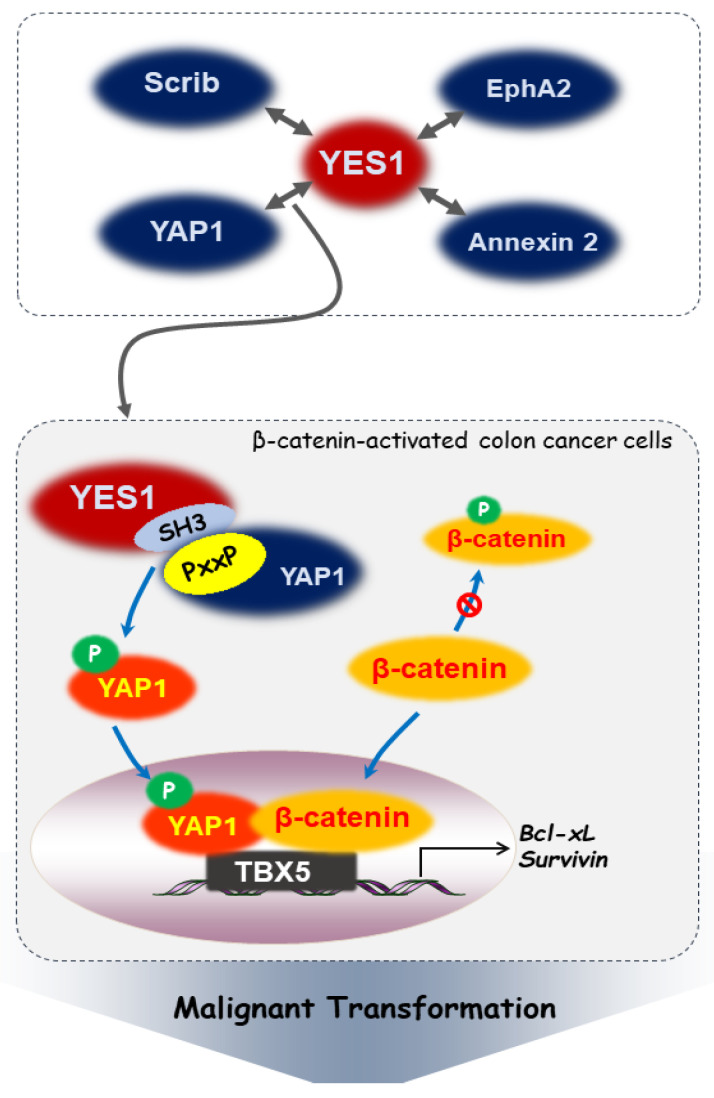
Proteins capable of interacting with YES1 and the corresponding signaling pathways. YES1 has been reported to regulate intracellular signaling by combining with Scrib, EphA2, annexin 2, and YAP1. Typically, unphosphorylated β-catenin avoids degradation, accumulates in the cytoplasm, and subsequently translocates to the nucleus. In colon cancer cells with activated β-catenin, when YES1 binds to YAP1, phosphorylated YAP1 enters the nucleus and forms a complex with β-catenin and TBX5, thus expressing apoptosis resistance proteins and promoting tumorigenesis.

## Data Availability

Not applicable.

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
