# Peer review of "Emerging Roles of YES1 in Cancer: The Putative Target in Drug Resistance"

_ijms, 2024, doi:10.3390/ijms25031450_

Round 1

Reviewer 1 Report

Comments and Suggestions for Authors

At present manuscript submitted by Kim et al., is concentrated on the YES1, one of Src family kinases (SFKs), which is a control tower in cancer focusing on drug-resistance. It is very intriguing that YES1 could present novel target molecular in many malignancies. However, the following questions should be conceived.

1.  The major: at present manuscript, the authors should be highlighted on the drug resistance of many malignancies. Please supplemented the relevant YES1 post translational modification of drug resistance. It is very importance for the manuscript.

2. Introduction: please supplemented the information or data for the presenting scientific hypothesis which means for the why and what?

3.  Why did the authors just concentrated on the YES1 activity? In fact, from the transcription, translation, and post translational modification of YES1 which maybe involve in the regulation of YES1.

4.  the part 4 Involvement of YES1 regulation in cancer and resistance to tumor therapy should be widely revised with the drug resistance of many malignancies.

5. Conclusion should be revised with the aim and contents. Especially, the authors should be presented the novel potentially beneficial strategies of malignancies. 

Comments on the Quality of English Language

The English language should be minor revised.

Author Response

First of all, the authors greatly appreciate the Associate Editor and reviewer’s general comments on the subject of the manuscript. These comments were very helpful in guiding us to prepare the revision more carefully and thoroughly. In accordance with Reviewer’s suggestion, we diligently revised manuscript. And, we revised the manuscript title to be more modest, in line with suggestions from other reviewers. We mark the added content in red.

  1. The major: at present manuscript, the authors should be highlighted on the drug resistance of many malignancies. Please supplemented the relevant YES1 post translational modification of drug resistance. It is very importance for the manuscript.

Reply) First of all, the authors greatly appreciate the reviewer’s comment. While revising, Sub-title No. 4 has been added, including general information about post-translational modification and details specific to YES1 (Page 4~5, Line 133~175). We mark the added content in red.

  1. Introduction: please supplemented the information or data for the presenting scientific hypothesis which means for the why and what?

Reply) We acknowledge the reviewer’s concern. Accordingly, we've written more background information at the beginning. It was changed according to the reviewer's suggestion to improve readability.

  1. Why did the authors just concentrated on the YES1 activity? In fact, from the transcription, translation, and post translational modification of YES1 which maybe involve in the regulation of YES1.

Reply) I really appreciate your comments on things I hadn't thought of, and I was able to organize them once again. Given that YES1 is a kinase, the title was originally crafted to emphasize its activity. Due to the inclusion of general information about transcription and translation, the title of subtitle 5 has been revised. The importance of such research is emphasized in the conclusion section.

  1. the part 4 Involvement of YES1 regulation in cancer and resistance to tumor therapy should be widely revised with the drug resistance of many malignancies.

Reply) We again acknowledge the reviewer’s concern. We changed the title of Part 5 and added more papers on non-coding RNA and resistance. In this review, we have tried our best to include all the papers reported so far. In addition, we tried to include in the conclusion the areas that require further research in the future.

  1. Conclusion should be revised with the aim and contents. Especially, the authors should be presented the novel potentially beneficial strategies of malignancies. 

Reply) We realized that our conclusion was somewhat limited and lacked clarity. We made changes based on the reviewer's suggestions. Thanks again to the reviewer.

  1. The English language should be minor revised.

Reply) Authors acknowledge the reviewer’s concern, so we carefully read this manuscript over. We have corrected the typographical errors in all. While performing revision, we had the proofreading done by a professional company.

Reviewer 2 Report

Comments and Suggestions for Authors

The review by Kook E. et al is devoted to the role of YES1 kinase in development of different kinds of cancer and its possible use as a target for antitumor drugs.  Also the mechanisms are considered by which expression and activation of YES1 are a potential factors contributing to the resistance to anti-cancer drugs. All this is important for the further development of anticancer therapy.

At the same time there are some comments to the authors. Considering the significance of YES1 activation  it is better to show its post-translational modifications as a visual diagram similar to Fig.2. Fig.2 demonstrate a good visual presentation of YES1 interaction with other proteins but

the right  pathway associated with β-catenin needs comments. A novel YES1 kinase inhibitor, called 341 CH6953755, with high specificity for YES1 is mentionedby the authors only in Conclusions while this compound has already exhibited responses in YES1-amplified esophageal and small-cell lung cancer (SCLC) xenograft models and should be more widely commented in the Section 4.3. And at last “ Try- 426 residue at line 129 should be changed on Tyr-426.

The review is interesting and important physiologically so it undoubtedly can be published.

Author Response

First of all, the authors greatly appreciate the Associate Editor and reviewer’s general comments on the subject of the manuscript. These comments were very helpful in guiding us to prepare the revision more carefully and thoroughly. In accordance with Reviewer’s suggestion, we diligently revised manuscript. And, we revised the manuscript title to be more modest, in line with suggestions from other reviewers. We mark the added content in red.

  1. Considering the significance of YES1 activation, it is better to show its post-translational modifications as a visual diagram similar to Fig.2. Fig.2 demonstrate a good visual presentation of YES1 interaction with other proteins but the right pathway associated with β-catenin needs comments.

Reply) We really appreciate the reviewers pointing out things we missed. We added more explanation to the figure section.

  1. A novel YES1 kinase inhibitor, called 341 CH6953755, with high specificity for YES1 is mentioned by the authors only in Conclusions while this compound has already exhibited responses in YES1-amplified esophageal and small-cell lung cancer (SCLC) xenograft models and should be more widely commented in the Section 4.3.

Reply) We are sincerely grateful to the reviewer for making suggestions to improve the readability of the content. We moved it around according to the reviewer's suggestion and modified the conclusion to make it more readable.

  1. And at last “ Try- 426 residue at line 129 should be changed on Tyr-426.

Reply) We thank the reviewer again for his thoughtful comments. We carefully read this manuscript over. We have corrected the typographical errors in all.

Reviewer 3 Report

Comments and Suggestions for Authors

In the current review, the authors summarized the current understanding about YES1 focusing on the regulation of YES1 and the involvement of YES1 in cancer development/treatment.

This review is well organized, and I only have few suggestions:

1. In the title the author pointed out "drug-resistance”. I would suggest the authors briefly introduce the current identified mechanisms involved in drug-resistance in cancer treatment, and summarize which mechanisms are related to YES1.

2. In the second section, the authors summarized several intracellular mechanisms involved in regulation of YES1. For my understanding, if the YES1 is defined as the “control tower”, I will expect to see the pathways and genes regulated by YES1 too.

3. Drug resistance is usually developed during the treatment. I wonder if there are any information about how YES1 regulated by anti-cancer drug treatment, in addition to intracellular molecules.

4. In the third section, “drug-resistance” was not seen for subtitle skin cancer, HCC, and glioma. If the authors want to emphasis the roles of YES1 in drug resistance, I would suggest the author to rewrite these sections.

Author Response

First of all, the authors greatly appreciate the Associate Editor and reviewer’s general comments on the subject of the manuscript. These comments were very helpful in guiding us to prepare the revision more carefully and thoroughly. In accordance with Reviewer’s suggestion, we diligently revised manuscript. And, we revised the manuscript title to be more modest, in line with suggestions from other reviewers. We mark the added content in red.

  1. In the title the author pointed out "drug-resistance”. I would suggest the authors briefly introduce the current identified mechanisms involved in drug-resistance in cancer treatment, and summarize which mechanisms are related to YES1.

Reply) We really appreciate the reviewers pointing out things we missed. While revising, we have added additional explanations at the beginning of Section 5. 

  1. In the second section, the authors summarized several intracellular mechanisms involved in regulation of YES1. For my understanding, if the YES1 is defined as the “control tower”, I will expect to see the pathways and genes regulated by YES1 too.

Reply) We thank the reviewer again for his thoughtful comments. We attempted to include all reported signaling related to the regulation of YES1 in section 3, and changed the title during revision to make it easier to understand. We revised the manuscript title to be more modest, in line with suggestions from other reviewers.

  1. Drug resistance is usually developed during the treatment. I wonder if there are any information about how YES1 regulated by anti-cancer drug treatment, in addition to intracellular molecules.

Reply) First of all, the authors greatly appreciate the reviewer’s comment. While revising, Sub-title No. 5 has been added, including general information about drug-resistance and details specific to YES1 (Page 5, Line 177~195). We mark the added content in red.

  1. In the third section, “drug-resistance” was not seen for subtitle skin cancer, HCC, and glioma. If the authors want to emphasis the roles of YES1 in drug resistance, I would suggest the author to rewrite these sections.

Reply) We thank the reviewer again for his thoughtful comments. We added a description under the title and subtitle to point out what we wanted to talk about in the section.